# Embodied Carbon Inventories for the Australian Built Environment: A Scoping Review

Josephine Vaughan *, Rebecca Evans and Willy Sher

School of Architecture and Built Environment, The University of Newcastle, Callaghan, NSW 2308, Australia; willy.sher@newcastle.edu.au (W.S.)

* Correspondence: josephine.vaughan@newcastle.edu.au; Tel.: +61-249-854-292

**Abstract:** Accounting for the embodied carbon in construction materials and calculating the carbon footprint of entire construction projects in life-cycle assessments is a rapidly developing area in the construction industry. Carbon emission accounting relies on inventories that claim to represent the values of carbon contained in materials. However, these values vary between different carbon inventories. This scoping review identifies academic research on the carbon inventories used in Australia, as well as the methods used to compare these inventories. The study was conducted in accordance with the JBI methodology for scoping reviews. We identified 182 papers and narrowed these down to 11 that complied with the objectives of this study. Data for a range of construction materials were compared in these papers, as were the methods used to calculate the values. While some carbon inventories were used frequently, no clear preference for the method of calculating carbon values was apparent. The system boundaries also varied between publications, and a range of functional units was used. There was agreement that the variables involved in calculating carbon values for building materials are compounded by the practical issues of extracting and manufacturing materials in different regional or local conditions, cultures, and technological situations. It is therefore understandable that different inventories store different values when so many factors need to be considered. There is thus a clear need for agreement to be reached about standardisation of the processes involved. If the trustworthiness of the data stored in carbon inventories is questionable, so too are the outcomes of subsequent activities.

**Keywords:** carbon emissions; construction materials; life-cycle assessment; material footprint; sustainable building





## 1. Introduction

Carbon accounting has become a world-wide priority. International, national, state, and local government bodies as well as many institutions and organisations have set embodied carbon targets as milestones. These targets are based on data which purport to represent the greenhouse gas emissions produced over the lifespan of products, commonly called *carbon emissions*. There are a variety of greenhouse gasses, of which carbon dioxide is one of the most significant. When accounting for greenhouse gas emissions, the impact of all gasses that contribute to the greenhouse effect are converted to an equivalent carbon dioxide value. When we discuss carbon equivalent emissions in this paper, we use the terms *carbon emissions*, *carbon values*, or *embodied carbon* for simplicity. These refer to all carbon equivalent gasses, which are often written as $CO_2e$ or $CO_2eq$. Throughout this paper, we have shortened *equivalent* to *eq* and have used $CO_2eq$.

In the built environment, the increase in building energy consumption is of concern [1,2]. While there are growing alternative energy sources producing renewable, clean energy and many new, innovative developments that increase energy efficiency and reduce operational carbon of buildings [3], in the construction industry, accounting for the embodied carbon in construction materials and calculating the carbon footprint of entire

construction projects is a rapidly developing area [4]. Embodied carbon emissions related to building materials must be considered to meet the United Nation's Sustainable Development Goals (SGDs) [5], specifically 'Affordable, Reliable, Sustainable and Modern Energy for All' (SDG7), 'Sustainable Cities and Communities' (SDG 11), and 'Responsible Consumption and Production' (SDG 12) [6]. With the construction sector contributing approximately 30% of carbon emissions worldwide [7], carbon reporting that was once optional is increasingly becoming mandatory. This is true not only for new buildings but for existing ones as well. International policies on zero carbon buildings have been adopted in many nations [8]. In Australia, BASIX (the compulsory NSW housing sustainability assessment) is presently being redesigned to include targets for the carbon embodied in construction materials [9]. Similarly, optional certifications for existing buildings, such as NABERS, are being developed to include carbon footprints [10].

Embodied carbon, carbon footprint, and life-cycle assessment (LCA) of individual construction projects may be conducted using in-house systems, consultancy services, or through subscribed or publicly available calculators [11]. The results of these assessments inform considerations of the impact of construction on climate change, as well as other sustainability criteria. All calculation approaches rely on inventories of the embodied carbon (or embodied energy) of individual construction materials. As described in the Climate Active Technical Guidance Manual, a carbon inventory is a "measure of the carbon dioxide equivalent emissions that are attributable to an activity. A carbon inventory can relate to the emissions of an [...] organisation, product, service, event, building or precinct. This can also be known as a carbon footprint or carbon account" [12].

However, there are over 20 carbon inventories for building materials worldwide. Some of the well-known ones include the Inventory of Carbon and Energy (ICE) Database [13] from the UK, GaBi [14] from Germany, and the ecoinvent Database [15] from Switzerland. There are at least six carbon inventories for building materials based in Australia, including the GreenBook [16] and EPiC [17].

The carbon values of construction materials can be represented in carbon inventories individually, such as concrete, brick, glass, timber, aluminium, and steel. The carbon values stored in inventories can also be provided for building system or assemblies. For example, window assemblies may include the carbon values of constituent materials, such as aluminium, glass, nylon thermal break, and plastic seals.

However, the values presented in different carbon inventories vary [18]. This is due to the wide range of methods used to manufacture construction materials as well as the methods used to determine their carbon value [19,20]. The production of construction materials varies from company to company. For example, for a brick company, there are differences between extracting and transporting raw materials as well as in manufacturing processes. This is true for each brick company, and for the different bricks produced by the company. The regional location where extraction and processing occurs can impact values as well, as do the prevailing culture and local practices [21]. Different manufacturing practices thus influence carbon emissions, as do travel distances between material producers and construction sites. Carbon inventories are also affected by the energy sources and processes available in different locations [22]. Any mismatch between the location of a project and the location inherent in a database can compromise the reliability of the data [21]. Furthermore, the methods (including the input–output method, the process method, and the hybrid approach that combines the two), the system boundaries (including the product stage (A1–A3); the construction process (A4–A5); the use stage (B1–B7); the end of life stage (C1–C4); and the benefits and loads stage beyond the system boundary(D)), [23] and other parameters (including functional units and reported values) used to calculate carbon content vary. Despite all these differences, carbon inventories often present the carbon values of different materials as if they were standardised. However, the embodied carbon value for any building material or system can be markedly different across different carbon inventories [24]. For example, the embodied carbon value for 1 m$^3$ of concrete may be 263 kgCO$_2$eq/m$^3$ in one carbon inventory (EPiC) [17] and 418 kgCO$_2$eq/m$^3$ in

another (ICE) [13]. Such different carbon values makes decision-making problematic. Stakeholders and governance bodies need to know if their embodied carbon predictions are accurate, meet benchmarks, or are comparable across different building projects [4,25,26]. Anomalies in these data undermine their credibility and compromise national carbon reduction statistics.

*Objectives*

The aim of this investigation was to understand the extent and types of evidence presented in academic, construction-related papers that compare different embodied carbon inventories. We sought to answer the question *What comparisons have been made in academic construction-related papers between different embodied carbon inventories in an Australian context?* A scoping review methodology was selected as this approach is ideally suited to identifying relevant exploratory research, the approaches others have used, and any gaps in knowledge [27]. Further details are provided below.

Whilst there is ample research about the tools that assess carbon emissions for buildings, we found few studies that compared the carbon inventories for individual construction materials. We therefore excluded comparisons between calculation tools and focussed on comparisons of carbon inventory data. To reiterate, our investigation deals specifically with the construction industry and is restricted to the construction of buildings or infrastructure in Australia.

This scoping review is part of a larger research project that aims to assess the authenticity of carbon inventory datasets insofar as they apply to the construction industry in Australia. This paper underpins the next stage of our work, which will involve quantitative comparisons of the embodied carbon of different construction materials across different databases. Therefore, this paper intentionally does not include such quantitative comparisons.

## 2. Materials and Methods

This scoping review was conducted in accordance with the JBI methodology for scoping reviews [28] and the PRISMA extension for scoping reviews [29]. Our research protocol was drafted using the JBI Manual for Evidence Synthesis [30]. The final protocol was registered with the Open Science Framework on 31 August 2023 (https://osf.io/jhak5, accessed on 14 February 2024). A preliminary search of the Open Science Framework, Figshare, and JBI Evidence Synthesis indicated that there were no current or ongoing systematic or scoping reviews on this topic.

This study focuses on carbon inventories of construction materials. Using a Population, Concept, Context (PCC) framework, the key elements used to conceptualise the review focus [31] are as follows:

Participants: Participants were not included in this review; this study is a review of published academic papers only.

Concept: This study relates to academic assessments of different carbon inventories for the built environment. Publications that do not consider the built environment were excluded. Similarly, publications that do not compare construction materials' carbon emission values were excluded.

Context: The construction industry in Australia.

*2.1. Search Strategy*

We searched for relevant studies published in refereed academic journals. An initial limited search of the Web of Science and Scopus identified pertinent articles. A full search strategy was developed from the titles and abstracts of these articles as well as the index terms used to describe them. The articles were then sourced from Scopus, Web of Science, and all databases within EBSCO Megafile Ultimate and Proquest (see Appendix A). The search terms were developed in conjunction with a research librarian at the University of Newcastle. The Science Direct database was trialled but not included as it did not cater for

all the search terms. As noted in Appendix A, the search strategy, including the identified keywords and index terms, was modified for each of the databases.

Only studies published in English were considered as it is unlikely that papers in other languages would be relevant to the Australian context. The time period investigated was between 1 January 2012 and 12 September 2022. This review considers academic articles, conference papers, reviews, data papers, books, and book chapters, but does not include company reports or product literature.

### 2.1.1. Selection of Evidence

Our initial search identified 182 papers (see Figure 1). These were collated, and duplicates were removed. The titles and abstracts of the remaining 114 papers were then reviewed against the inclusion criteria. This resulted in 90 papers being discarded. The full text of the remaining 24 papers was then retrieved and read in full. This resulted in a further 13 papers being discarded for the reasons noted in Figure 1. The remaining 11 papers constitute the data which form the basis of this scoping review. All of these papers compared the embodied carbon values of individual building materials (and assemblies of materials) stored in different carbon inventory databases.

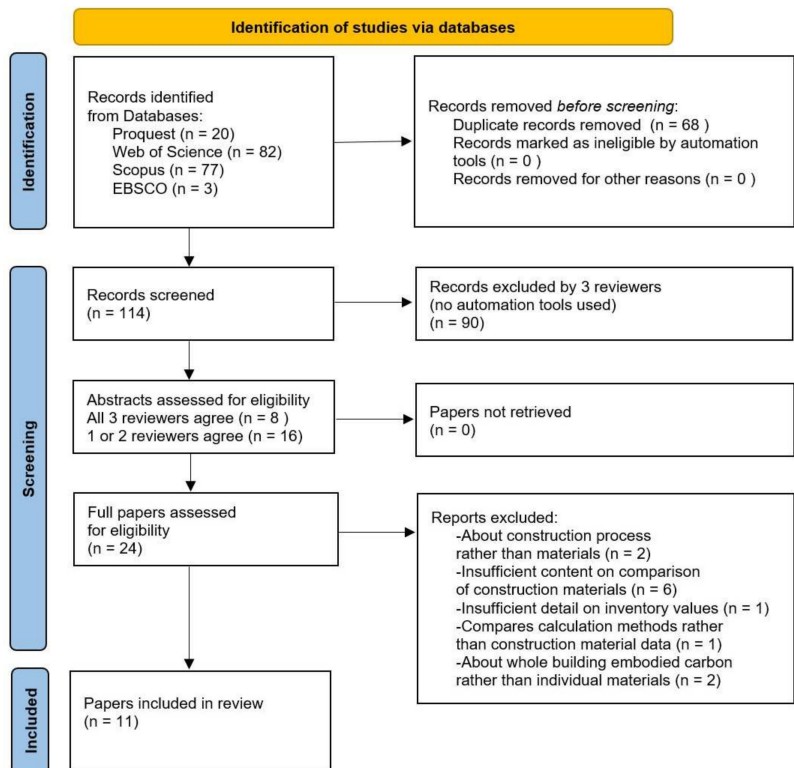

**Figure 1.** Selection of publications based on PRISMA flow diagram [32].

### 2.1.2. Extraction of Evidence

Data were retrieved using an extraction tool developed during the scoping review protocol process (see Appendix B). An extraction form was designed to include specific details about the concept, context, study methods, and key findings relevant to the review question (*What comparisons have been made in academic construction-related papers between different embodied carbon inventories in an Australian context?*). The form allowed the following to be recorded: details of the carbon inventories compared (which inventories, carbon calculation type); units of consideration used in comparisons (carbon or energy, system boundaries, functional units, values compared); how the study was conducted (method of comparison, materials studied, comparison focus, what was compared); the results and key findings; and any other discussion relevant to the research question.

The reviewers mostly worked independently to enter data into the form but discussed nuances with co-investigators when necessary. The Chief Investigator checked the data for consistency. The data are presented below. A narrative summary accompanies the tabulated results and describes how the results relate to the question addressed by this review.

## 3. Results

### 3.1. Review of Sources

All 11 studies compared the embodied carbon values of individual building materials (and/or assemblies of materials) stored in different carbon inventory databases. These papers were published at a steady rate between 2014–2022 but more (four) were published in 2020. In any specific and geographically limited research space, it is typical that authors publish more than one document on the topic and may also collaborate with each other. In the set of literature selected, 9 of the 11 papers had crossover work by other authors in the group. There were various sources of carbon values in each study. Some studies generated their own carbon values using various calculation methods, others applied carbon values from what we called 'branded' databases, such as EPiC Database [17], while some applied independently calculated values presented in other publications. In addition, others aggregated or compiled new values from any of those sources. Many papers included a combination of carbon value sources (Table 1).

**Table 1.** Characteristics of sources of evidence.

| Publication | Year | Some or All Authors Included in Other Publication(s) | Study Draws on a 'Branded' Database for Carbon Values | Study Draws on Academic Publications for Carbon Values | Study Includes Values: Generated for the Publication (G); or Aggregations of Other Values (A); or Compilations for Composite Materials (C); or Directly Using 'Branded' Inventory Data (D) |
|---|---|---|---|---|---|
| Wan Omer et al. [33] | 2014 | | | • | G, D |
| Robati et al. [34] | 2016 | • | • | • | C, D |
| Teh et al. [35] | 2017 | • | | • | G, D |
| Teh et al. [36] | 2018 | • | | • | G, D |
| Crawford et al. [37] | 2019 | • | • | • | D |
| Robati et al. [38] | 2019 | • | • | • | A |
| Helal et al. [39] | 2020 | • | • | | D |
| Allende et al. [40] | 2020 | • | • | | D |
| Crawford & Stephan [41] | 2020 | • | • | | D |
| Rodrigo et al. [18] | 2021 | | • | | G |
| Robati & Oldfield [42] | 2022 | • | • | • | A |

### 3.2. Characteristics of Sources

To identify how the sources compared carbon values of different building materials in one inventory with those in a different inventory, we investigated five overarching characteristics of the selected publications: (i) their focus; (ii) the materials considered in the publications; (iii) carbon value inventories; (iv) calculation approaches; and (v) results. These are illustrated in Table 1 and explored in more detail below.

### 3.2.1. Focus Areas of Publications

All 11 studies compared the embodied carbon values of individual building materials (and/or assemblies of materials) stored in different carbon inventory databases. While there were studies that included comparisons of databases within Australia, no papers had the sole objective of comparing the carbon database differences within Australia. Three of the papers included specific comparisons of the carbon values stored in databases, including systematised comparisons of carbon inventory data stored in different carbon inventory databases. In eight of the studies, comparisons were reported as part of a larger study rather than being the main focus of the research. Of these eight, six emphasized LCI methodologies and/or the calculation of carbon coefficients; two focussed on the relationship of $CO_2$eq to material properties or types; and another two investigated areas of uncertainty in calculating embodied carbon for a whole building (Table 2).

**Table 2.** Focus areas of publications.

| Publication | Did Publication Specifically Compare Databases or Was This Part of a Larger Study? | | | |
|---|---|---|---|---|
| | Yes—Specifically about Comparing Databases | No—About LCI Methodologies and/or Inputs (the Calculation of Carbon Coefficients) | No—About Relationship of $CO_2$eq to Material Properties or Types | No—About Areas of Uncertainty in Calculating Embodied Carbon for a Whole Building |
| Wan Omer et al., 2014 [33] | | • | | |
| Robati et al., 2016 [34] | | | • | |
| Teh et al., 2017 [35] | | • | | |
| Teh et al., 2018 [36] | | • | | |
| Crawford et al., 2019 [37] | | • | | |
| Robati et al., 2019 [38] | | | | • |
| Helal et al., 2020 [39] | | • | | |
| Allende et al., 2020 [40] | • | | | |
| Crawford & Stephan 2020 [41] | • | | | |
| Rodrigo et al., 2021 [18] | • | • | | |
| Robati & Oldfield 2022 [42] | | | • | • |

In addition to the foci shown in Table 2, other themes were apparent in the publications. All explored the values of carbon data. Of those that specifically compared databases, Allende et al. [40] and Crawford & Stephan [41] investigated the effect of data age on carbon values. In their study, Rodrigo et al. [18] developed a data set/methodology (SCEEM); they compared their results to the Blackbook database as well as values provided by eToolLCD (eToolLCD is not a carbon inventory; it is a carbon calculator that relies on other inventories). Other investigators either examined variations in data more generally, explored uncertainty in the development of a new methodology/data set, or investigated the causes of variations in carbon data [33,34,38,42]. Other comparisons were included in the work of Allende et al. [40] and Crawford and Stephan [41], who reviewed the impact of data age in hybrid data sets. Four studies looked at the effect of life-cycle assessment (LCA) on carbon values. Teh et al. [35,36], Helal et al. [39] and Crawford et al. [37] all compared process LCA data to hybrid LCA data and one study [1] also compared "environmentally extended input–output" data.

### 3.2.2. Materials Considered

While all publications provided values for individual materials (for example, glass), five also included calculations for assembled products (such as windows), six provided comparisons of individual materials or assemblies and combined individual material data in case studies to compare the carbon values in or of whole buildings, or square metre rates for buildings. Some included calculations for a whole building and Allende et al. [40] took this further by making comparisons across a 14,000 m$^2$ development. Only one publication included carbon values for traditional construction processes [18] (Table 3).

**Table 3.** Construction components included in study.

| Publication | Components Studied | | | |
|---|---|---|---|---|
| | Individual Material (e.g., Glass) | Assembled Product (e.g., Window) | Entire Building | Construction Process |
| Wan Omer et al., 2014 [33] | • | | | |
| Robati et al., 2016 [34] | • | | | |
| Teh et al., 2017 [35] | • | | | |
| Teh et al., 2018 [36] | • | | | |
| Crawford et al., 2019 [37] | • | | • | |
| Robati et al., 2019 [38] | • | • | • | |
| Helal et al., 2020 [39] | • | • | • | |
| Allende et al., 2020 [40] | • | • | • | |
| Crawford & Stephan 2020 [41] | • | • | • | |
| Rodrigo et al., 2021 [18] | • | | | • |
| Robati & Oldfield 2022 [42] | • | • | • | |

A representative sample of materials commonly used in domestic construction was apparent in the papers, including aluminium, brass, bricks, blocks, carpet, cement, ceramic tiles, concrete, copper, fibre cement, glass, insulation, mortar, paint, pavers, plasterboard, plastics, roof tiles, sand, steel, stone, synthetic rubber, and timber. Data pertaining to

these materials were recorded in several databases, including EPiC [17], ICE [13], and AusLCI [43].

Five publications investigated more than two individual materials [33,38,40–42]. The others focused on only one or two, typically concrete [34–37,39] (Please see details in Appendix C). Concrete was most frequently studied (n = 9), followed by steel (n = 6) (Figure 2).

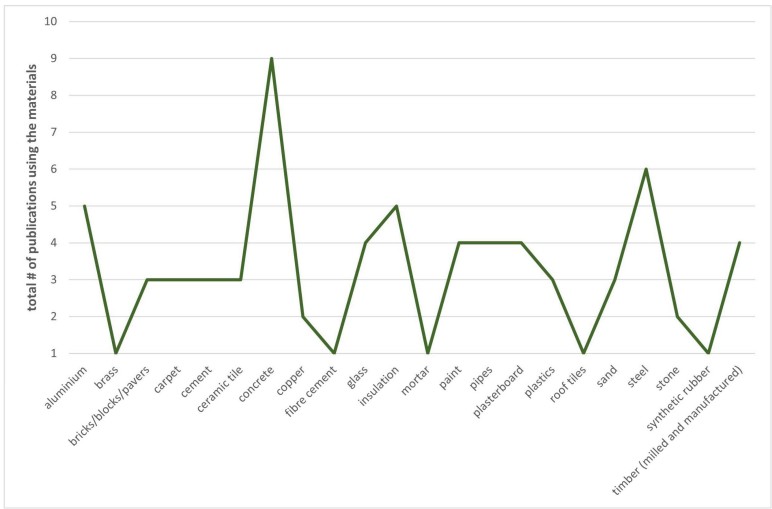

**Figure 2.** Construction materials included in publications.

Concrete elements (such as structural systems that included reinforcing) were investigated in the publications (n = 3). Other assemblies of materials examined in multiple studies were envelope elements, such as external walls (n = 3), internal walls (n = 3), roof systems (n = 3), and openings (doors n = 2, windows n = 3) (Figure 3).

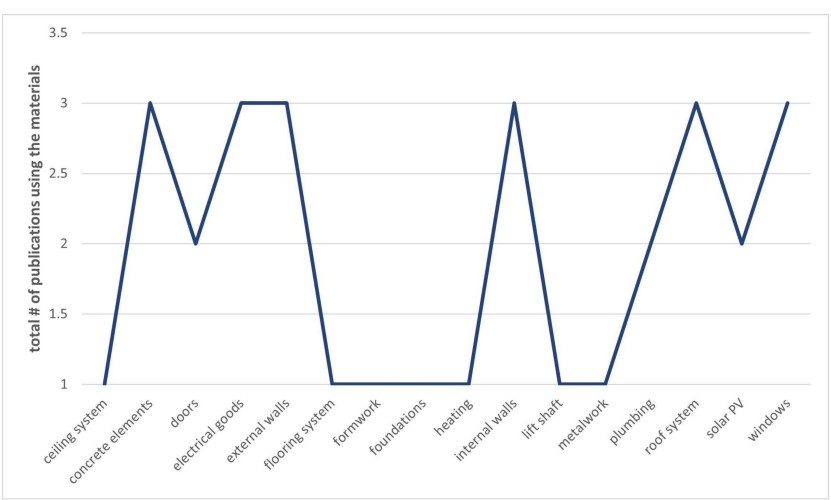

**Figure 3.** Assembled materials studied in publications.

### 3.2.3. Inventories of Carbon Values

The carbon values recorded in the publications were from a range of sources. As shown in Table 4, all drew on more than one source. For clarity, the sources of carbon value data were grouped into three main areas: (a) self-generated; (b) from existing carbon inventories; or (c) from existing carbon publications. Of the four papers that generated their own carbon values, the base (upstream) data used were from Environmental Product Declarations (EPDs), existing carbon inventories, or other unspecified data. These papers also presented carbon values from other sources, usually to compare to the carbon values they had generated themselves.

**Table 4.** Sources of carbon values used in publications.

| Publication | Source of Carbon Values | Wan Omer et al., 2014 [33] | Robati et al., 2016 [34] | Teh et al., 2017 [35] | Teh et al., 2018 [36] | Crawford et al., 2019 [37] | Robati et al., 2019 [38] | Helal et al., 2020 [39] | Allende et al., 2020 [40] | Crawford & Stephan 2020 [41] | Rodrigo et al., 2021 [18] | Robati & Oldfield 2022 [42] |
|---|---|---|---|---|---|---|---|---|---|---|---|---|
| (a) Self-Generates carbon values using an inventory methodology presented within publication | AusLCI | | | • | • | | | | | | | |
| | ecoinvent | | | | • | | | | | | | |
| | EPDs | • | | | | | | | | | • | |
| | ICE 2.0 | • | | | | | | | | | | |
| | IELab | | | • | • | | | | | | | |
| | Other data (no database given) | • | | • | • | | | | | | • | |
| (b) Directly uses, compiles or aggregates values from 'branded' inventory | AusLCI | | • | | | | • | | | | | |
| | Blackbook | | | | | | | | | | • | |
| | BPIC | | • | | | | • | | | | | • |
| | Database of EE & water values | | | | | | | | • | • | | |
| | EPiC | | | | | • | | • | • | • | | • |
| | EtoolLCD * | | • | | | | • | | | | • | |
| | ICE | | • | | | | • | • | | | | • |
| | ICM | | | | | | | | | | | • |
| (c) Directly uses, compiles or aggregates direct values derived from academic publication | ADAA, 2016 as cited in [34] | | • | | | | | | | | | |
| | Alcorn, 2003 as cited in [33,34,38,42] | • | • | | | | • | | | | | • |
| | Chen et al., 2010 as cited in [42] | | | | | | | | | | | • |
| | Crawford, 2011 as cited in [33,34,38,42] | • | • | | | | • | | | | | • |
| | Davidovits, 2015 as cited in [35,36] | | | • | • | | | | | | | |
| | Flower and Sanjayan, 2007 as cited in [34] | | • | | | | | | | | | |

**Table 4.** *Cont.*

| Publication | Source of Carbon Values | Wan Omer et al., 2014 [33] | Robati et al., 2016 [34] | Teh et al., 2017 [35] | Teh et al., 2018 [36] | Crawford et al., 2019 [37] | Robati et al., 2019 [38] | Helal et al., 2020 [39] | Allende et al., 2020 [40] | Crawford & Stephan 2020 [41] | Rodrigo et al., 2021 [18] | Robati & Oldfield 2022 [42] |
|---|---|---|---|---|---|---|---|---|---|---|---|---|
| (c) Directly uses, compiles or aggregates direct values derived from academic publication | Grant, 2015 as cited in [35] | | | • | | | | | | | | |
| | McRobert, 2010 as cited in [34] | | • | | | | | | | | | |
| | Moussavi Nadoushani and Akbarnezhad, 2015 as cited in [38,42] | | | | | | • | | | | | • |
| | Pullen, 2007 as cited in [33] | • | | | | | | | | | | |
| | Robati et al., 2016 as cited in [38,42] | | | | | | • | | | | | • |
| | Robati et al., 2019 as cited in [42] | | | | | | | | | | | • |
| | Rouwette, 2012 as cited in [34] | | • | | | | | | | | | |
| | Teh et al., 2017 as cited in [36,42] | | | | • | | | | | | | • |
| | Teh, 2018 as cited in [42] | | | | | | | | | | | • |
| | Turner et al., 2013 as cited in [35,36] | | | • | • | | | | | | | |
| (d) Source not clearly explained | "process data" | | | | | • | | | | | | |

* eToolLCD is a carbon calculator that relies on other inventories.

The remaining seven papers (shown in sections b, c, and d of Table 4) either directly included existing carbon values or aggregated or compiled carbon values directly from branded carbon inventories or from academic publications.

All publications used a branded carbon inventory either as a data source, used it for general values, or used it for both. A total of nine branded databases were used across the 11 papers (Figure 4). The most frequently used were EPiC (n = 5) and the Inventory of Carbon and Energy (ICE) (n = 5), followed by AusLCI (n = 4). EPiC and Aus LCI are Australian-based carbon inventories whilst ICE is based in UK. ICE is a free access database that has been available since 2005 and is used internationally as a source of carbon values and as an academic reference point [44].

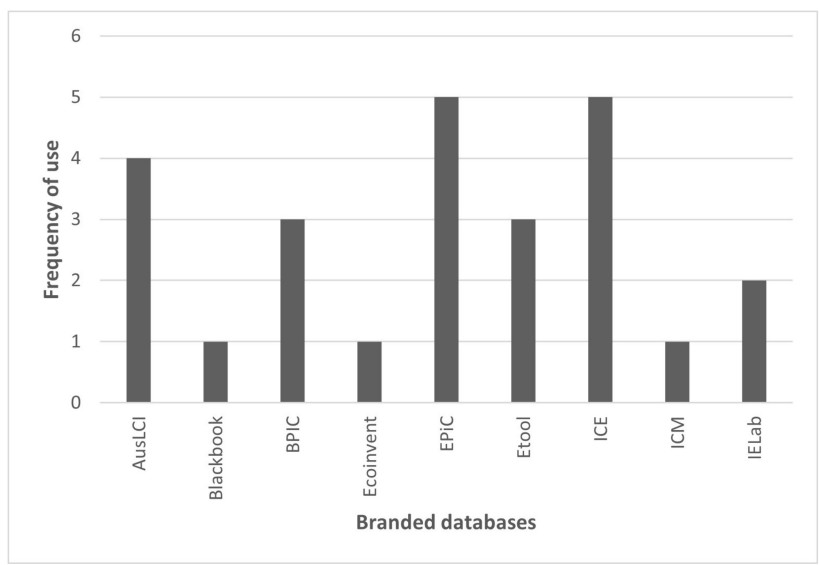

**Figure 4.** Frequency of carbon inventory use in the sources.

3.2.4. Calculation Approaches

There are three main approaches to calculating carbon values. The *process* methodology (PLCA) is also known as a *bottom-up* approach where the carbon emissions of all aspects of producing an individual material, from extraction onwards, are calculated according to ISO standards [4]. EPDs are prepared using the process calculation approach, following the BS EN 15804:2012+A2:2019 standard [45]. The process methodology is very detailed and is a very reliable method of assessing embodied carbon [19]. However, to achieve this, the level and extent of information required means that it is very time consuming and generally requires experts to calculate the embodied carbon values [46]. The second approach, the *input–output* methodology (IOLCA), is also known as the *top-down* approach. Here, economic data for industries is aligned with carbon intensity, and generalised material carbon values are produced [47]. The advantage of the input–output method is that it provides localised data at a national level. However, these data are very broad and, thus, do not address the peculiarities of individual materials, meaning that assumptions are made about much of the data. Furthermore, some of these data may be out of date by the time they are applied [46]. Third, the *hybrid* methodology (HLCA), also known as the *combined* approach, brings together both process and input–output data to collect as many upstream data points as possible to calculate a carbon value [48]. While this method is faster than the process method and more accurate than the input–output method, its accuracy is compromised as it relies on the assumptions made during the input–output approach [46].

Regarding the selected publications, the three led by Robati [34,38,42] did not clearly specify which method or methods were used in their studies. Helal et al. [39] and Wan Omer et al. [33] applied all three methods, while Teh et al. [35,36] and Crawford et al. [37] used both process and hybrid methods separately. Of the other papers, Allende et al. [40]

and Crawford and Stephan [41] only used the hybrid approach, and Rodrigo et al. [18] applied the process method only (Table 5).

**Table 5.** Carbon value calculation methodologies used in the publications.

| Reference | Carbon Value Calculation Type | | | |
|---|---|---|---|---|
| | **Process** | **Input-Output** | **Hybrid** | **Not Specified** |
| Wan Omer et al., 2014 [33] | ● | ● | ● | |
| Robati et al., 2016 [34] | | | | ● |
| Teh et al., 2017 [35] | ● | | ● | |
| Teh et al., 2018 [36] | ● | | ● | |
| Crawford et al., 2019 [37] | ● | | ● | |
| Robati et al., 2019 [38] | | | | ● |
| Helal et al., 2020 [39] | ● | ● | ● | |
| Allende et al., 2020 [40] | | | ● | |
| Crawford & Stephan 2020 [41] | | | ● | |
| Rodrigo et al., 2021 [18] | ● | | | |
| Robati & Oldfield 2022 [42] | | | | ● |

Different variables need to be considered when the carbon values of building materials are assessed (Table 6). These include the life-cycle stages included in any calculation of embodied carbon values. These stages are commonly referred to as the system boundaries, as defined by the British Standards Institution [23]. The boundaries in the reviewed papers varied. Three used A1–A3 (raw material supply to manufacturing), five used A1–A5 (raw material to construction–installation process), three also considered stage B (the operational stage of a building), and two included stage C (the end-of-life stage). Two publications did not specify the stages considered.

**Table 6.** Units of consideration in the publications.

| Publication | Units of Consideration | | |
|---|---|---|---|
| | **System Boundary** | **Functional Unit** | **Values Compared** |
| Wan Omer et al., 2014 [33] | not specified | kg | MJ/kg<br>$CO_2$eq/kg |
| Robati et al., 2016 [34] | A1–A3 | $m^3$ | kg $CO_2$eq/$m^3$ |
| Teh et al., 2017 [35] | A1–A3 | kg, $m^3$ | kg $CO_2$eq/$m^3$<br>kg $CO_2$eq/kg |
| Teh et al., 2018 [36] | A1–A3 | kg, $m^3$ | kg $CO_2$eq/$m^3$<br>kg $CO_2$eq/kg |
| Crawford et al., 2019 [37] | A1–A5, B1, B5, B6 | $m^3$ | GJ/element<br>Gj/whole building |
| Robati et al., 2019 [38] | A1–C1 | #, kg, $m^2$, $m^3$ | kg $CO_2$eq/$m^2$<br>kg $CO_2$eq/$m^3$<br>kg $CO_2$eq/kg<br>kg $CO_2$eq/# |
| Helal et al., 2020 [39] | A1–A5 | kg, $m^2$ | kg $CO_2$eq/$m^2$<br>kg $CO_2$eq/kg<br>GJ/$m^2$ |
| Allende et al., 2020 [40] | A1–A5, B4 | $m^2$ | kg$CO_2$eq/$m^2$<br>GJ/material<br>kg$CO_2$eq/material<br>GJ/assembly<br>kg$CO_2$eq/assembly |
| Crawford & Stephan 2020 [41] | not specified | #, kg, $m^2$, $m^3$ | GJ/material<br>GJ/element<br>GJ/$m^2$ |
| Rodrigo et al., 2021 [18] | A1–A5 | $m^2$, $m^3$ | kg $CO_2$eq/$m^2$<br>kg $CO_2$eq/$m^3$ |
| Robati & Oldfield 2022 [42] | A1–A5, B1, B4, C1, C3, C4 | t, m, $m^2$, $m^3$ | kg $CO_2$eq/$m^3$<br>(kg $CO_2$eq/$m^3$)/(W/mK) |

Embodied carbon calculations also include a functional unit which represents a comparable amount of construction material or building project. These can very between assessments, adding an additional variable when attempting to compare values [8], and can be measured in length, area, volume, number, weight, or any other given comparison point. The functional units presented in the publications varied. They were mostly based on whether the study investigated materials, elements, or buildings. They included kg, t, $m^2$, $m^3$, and total buildings and multi building developments (Table 6). Some studies included discussions and/or comparisons of the actual functional unit of materials in relation to their data source co-efficient. For example, Teh et al. [36] compared and presented multiple EE or $CO_2eq$ co-efficients against a functional unit of a material. Other publications focused on calculation methodologies and referred to the co-efficients more generically. For example, Crawford et al. [37] reported multiplying the item quantities in a bill of quantities by EE or $CO_2eq$ co-efficients. They then discussed the total outcome in relation to an entire building. Some studies, such as Robati and Oldfield [42], reported on multiple functional units, such as $m^3$ for material and $m^2$ for the whole building. Others, such as Allende et al. [40], reported on the total emissions related to a material or assembly within a building to understand the impact at a whole building level.

The impact of a building material on carbon emissions was typically measured as the carbon emissions expressed as kilograms of carbon dioxide equivalent ($kgCO_2eq$) per amount of material. Some also included the amount of embodied energy, expressed as Megajoules (MJ) or Gigajoules (GJ) per amount of material. Only two papers did not include carbon emission values and only presented measures of embodied energy. Seven papers only presented embodied carbon values, and the remaining two papers presented both.

### 3.2.5. Presentation of Results

Most of the results in the studies were presented as bar charts and tables (Figure 5). In total, 7 of the 11 sources used more than one form of visualisation (see Appendix D).

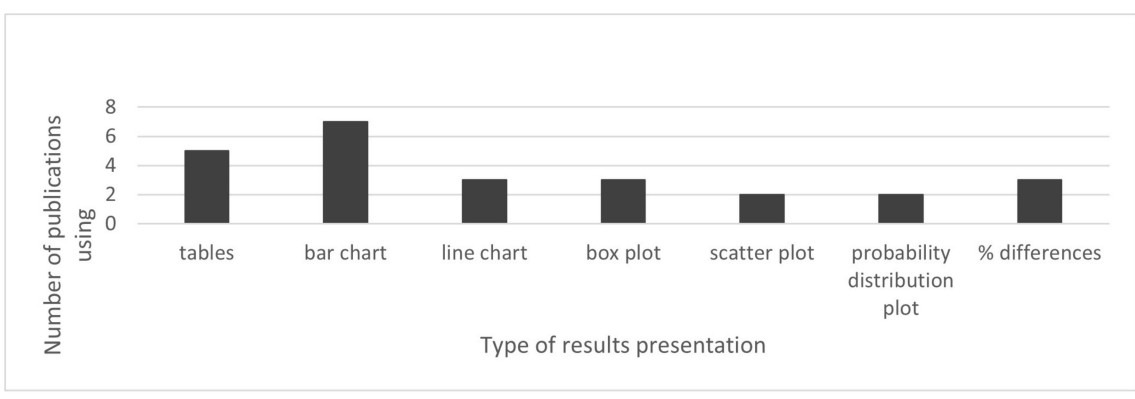

**Figure 5.** Frequency and type of results presentation.

While each paper focussed on a discrete topic and thus had a unique set of findings, there were some similar themes. All of the publications noted that there were significant variations in the embodied carbon values for the same building materials, across different carbon inventories. Eight publications noted that these were caused by different factors used in calculations, including for extraction and manufacturing technology, energy/fuel, material price, source of raw materials, transportation, climate, and also changes in any of these over time [33,34,36,37,40–42]. Variations were also identified as being caused by over and/or under estimations in calculation methods. For example, calculations made using data collected and combined through process, input–output, and hybrid methods, or generated by an inconsistent mix of calculation methods, resulted in these variations [34–37,39]. Another cause of variations was identified as inconsistent upstream data or data components that were missing or estimated. This was particularly the case for multi-ingredient materials (such as glass, plasterboard, insulation, and concrete mixes) [18,33–35,38–40,42].

The use of different, inconsistent, incomplete system boundaries was also identified as a cause of variations [34,37,42].

Some publications noted that, despite variations in carbon values, some embodied carbon values for discrete materials were consistent across some carbon inventories. This was the case when a greater percentage of process data was included for materials, particularly where direct energy was used in the process, such as in steel, cement, and brick. The authors argued that this provided more realistic results [33,37,41]. Likewise, it was identified that good quality data used from the outset results in more consistent embodied carbon values [36].

Improvements in ongoing carbon emission data collection were suggested in some publications. These include transparency of calculations [34,39] improving the quality of the data [35], and using nation-specific data [36].

## 4. Discussion

### 4.1. The Embodied Carbon Problem

This paper stems from an observation that different carbon inventories store different carbon values for similar construction materials. A preliminary literature review confirmed this and highlighted the concerns of researchers in several countries [26,49–51]. Some of the reasons for these differences have been noted earlier. They include different extraction, transportation, and manufacturing methods as well as local and regional practices, the time when processes occurred, energy sources, and the methods used to calculate the carbon values. Importantly, there is no standardisation, governance, or regulation of the calculation and collection of embodied carbon values of building materials. While international standards (ISO 14040: 2006, ISO 14044: 2006, EN 15978:2011 and EN 15804:2012 [23,45,52,53]) are generally complied with, the literature emphasises that these allow for broad interpretation and result in inconsistent carbon values [19,54]. No single international or Australian national body is responsible for auditing, accrediting, and hosting a standardised database. This extends to regulating the calculations and collections of data for embodied carbon values. Despite this, the construction industry appears keen to move forward [55]. As mentioned earlier, BASIX is being redesigned to include carbon targets for construction materials [9]. Similar sustainability assessment systems are in operation in other states, and it is likely they will follow NSW in setting carbon reduction targets. Furthermore, several independent databases have come to the international and national market since the early 2000s, and new ones continue to become available, including the recently released Building Environment Carbon Database in the UK [56].

Additionally, existing industry groups have created their own embodied carbon standards. For example, the Royal Institute for Charters Surveyors (RICS) has created a professional standard for their members [57]. The need for leadership and consistency has given rise to industry-led national collaborations, such as the Materials and Embodied Carbon Leaders' Alliance (MECLA) in Australia. This organisation is working to develop solutions for the construction industry to reduce embodied carbon. Although the reasons for the differences in carbon values presented in different inventories are understood, the practicalities of addressing them remain. Sourcing and collating accurate data from such a wide array of sources is understandably challenging. Concern about this motivated the investigation to identify comparisons of different embodied carbon inventories presented in academic construction-related literature in an Australian context.

### 4.2. Discussion of Results

A scoping review methodology was adopted for this research to pave the way for future studies of the carbon inventories used in Australia. Our initial searches unearthed 182 papers. After screening for relevance and duplicates, only 11 were identified as applicable. This modest number indicates that either the area is under-researched or the differences are acknowledged and, perhaps, accepted [58]. Neither explanation is satisfactory.

Most of the authors of the selected papers described the various methodologies they used to assess the environmental impact of construction materials, including LCI and $CO_2$ eq. Whilst these approaches are incidental to the aim of this paper, they illustrate the need to explain how carbon data are calculated and, in some cases, how they were used. Indeed, the calculation of carbon values is complex. A recent evaluation of carbon in construction materials identified 143 standards that apply at some point in the calculation process [24]. It is therefore unsurprising that practitioners as well as laypeople do not appreciate the complexities involved.

Comparisons of carbon data for a range of construction materials were presented in the papers. Concrete was most frequently studied, followed by steel. Various elements and assemblies of materials were also considered, with those made from concrete being the most explored. However, whilst knowledge and understanding of these materials has grown, some other popular materials remain under investigation. For example, only three publications investigated plastics. Fibre cement and roof tiles attracted a single study each. The abundance of different materials used in buildings presents obvious challenges for researchers. The basis on which materials are selected for investigation is opaque. The conundrum facing researchers is whether to study materials that are currently sparsely researched or continue to investigate popular materials (i.e., those with significant carbon content and/or the potential for them to be reused and/or recycled).

A total of nine branded databases were used across the 11 papers. EPiC [17] and the Inventory of Carbon and Energy (ICE) [13] were the most popular databases. As mentioned previously, EPiC is an Australian-based carbon inventory, whilst ICE is based in the UK. When a database from a different location is used, researchers need to allow for differences between locations. This is generally achieved by applying factors that compensate for these differences. For example, to allow for the manner in which power is generated, transport distances and preferences, and so on. Clearly, it is logical to use an Australian database for projects in Australia.

The majority of the papers compared embodied carbon values between different databases. These were presented as part of their study into the method of calculating carbon. Whilst no clear preference for a particular method (i.e., the process method, the input–output method, or the hybrid method) was expressed in the 11 papers, most either used the input–output or hybrid method. Despite claims by some authors that the process method provided the most realistic results, this approach received the least attention in calculation methodology discussions. It was noted that the different methods of calculation produce different values. This is understandably concerning when comparisons of carbon values are made. However, the calculation method is not the only variable that impacts a carbon value. The authors of some papers used different system boundaries in their calculations, and these also affected their comparisons, as complicated equivalencies were required to compare data. It is worth noting that these system boundaries often vary for the values presented in carbon inventories. Amongst the papers, 27% used the system boundaries of A1–A3 and 45% used A1–A5. Stage B was considered by 27% and stage C by 18%. Finally, when numerical values are compared, the units of measurement for each item need to be the same. However, due to the nature of individual materials and construction practices, it makes sense for some material carbon values to be presented by weight, others by volume, and still others by area. This variation of units is common in industry practice (such as quantity surveying) and prevails in carbon inventories. Likewise, the functional units presented in the publications varied and included kg, t, $m^2$, $m^3$, and total buildings and multi building developments.

On a positive note, there has been a move across carbon inventories to consistently report embodied carbon (kg$CO_2$ eq) of building materials rather than the previous practice of reporting either embodied energy (MJ or GJ) or embodied carbon, or both [44]. The more recent practice of providing data as embodied carbon (kg$CO_2$ eq) was reflected in the sources, with all but two papers including carbon values.

The authors of the documents reviewed for this study emphasised that the significant variables involved in calculating carbon values for building materials are compounded by the practical issues of extracting and manufacturing materials in different local conditions, regions, cultures, and technological situations. These result in a further range of possible variables and, thus, values for a building material. It is thus understandable that different inventories store different values when so many factors need to be allowed for.

In summary, some data are available to facilitate an investigation of the differences in carbon values for some materials. In the next phase of our work, we will explore the values stored in various databases and evaluate how different they are.

## 5. Conclusions

This scoping study has investigated published comparisons of the carbon inventories used in Australia. We also explored the research methods used to investigate these inventories. Whilst our initial search unearthed 182 papers, only 11 were considered relevant. The authors of these papers agreed that there were inconsistencies in the values presented in the inventories. Several reasons were noted, including different methods of calculation, system boundaries, and functional units. Other reasons included different extraction, transportation, and manufacturing methods as well as local and regional practices, the time when processes occurred, and energy sources. Bearing all of these in mind, it would be surprising if different inventories did present similar values. There is thus a clear need for agreement to be reached about standardisation of the processes involved. Some progress has been made by the Materials Embodied Carbon Leaders' Alliance (MECLA) in Australia. Their recent report documents current developments about upfront embodied carbon standards, measurement, benchmarking, and resources for construction materials, buildings, and infrastructure. The first step in mapping the landscape of carbon accounting in Australia should inform future work in defining standards. However, standardisation itself is not a panacea, and there is a need for strong governance, leadership, and responsibility of carbon data management to be enacted.

This lack of reliability gives rise to several questions which we intend to address in future studies. For example, how different are the data sets used in Australia? Is there a statistically significant difference between them? If there is a difference, what impact does this have? Could the outcomes from various calculators be harmonized to generate a result across several data sets? Would such a harmonized result reflect reality? These questions all rely on carbon inventory data that accurately reflect the carbon emissions of the materials in question. If the trustworthiness of these data is questionable, so too are the outcomes of subsequent activities.

*Limitations*

This paper has focussed exclusively on academic literature that compares carbon data for construction materials used in Australia. A scoping review was conducted on papers published between 1 January 2012 and 12 September 2022. A scoping perspective was purposively selected as the outcomes of the study are intended to inform and steer continuing work on the topic. Articles that complied with the selection criteria were sourced from Scopus, Web of Science, and all databases within EBSCO Megafile Ultimate and Proquest. These databases provide comprehensive coverage of the sources of academic articles on the identified topic. Three reviewers participated in identifying and reviewing material. The chief investigator provided quality assurance of the entire process by reviewing the assessments and coding their colleagues' work.

**Author Contributions:** Conceptualization, J.V., R.E. and W.S.; methodology, J.V. and W.S.; validation, J.V. and R.E.; formal analysis, J.V. and R.E.; data curation, J.V., R.E. and W.S.; writing—original draft preparation, J.V. and W.S.; writing—review and editing, J.V., R.E. and W.S.; visualization, J.V.; supervision, J.V. and W.S.; project administration, J.V.; funding acquisition, J.V. All authors have read and agreed to the published version of the manuscript.

**Funding:** This research was funded by the Footprint Company [grant number CESE IMF 22] and the University of Newcastle [grant number G2200888].

**Data Availability Statement:** Not Applicable.

**Acknowledgments:** Caroline Noller provided advice for the conceptualization and review.

**Conflicts of Interest:** The research is funded by the Footprint Company which is a business offering a carbon calculation tool and embodied carbon database. The research was initiated by the Footprint Company to ensure they are taking valid and accurate approaches to using carbon inventories.

## Appendix A. Search Strategy

The search terms used for all databases were:

"embodied carbon" OR "carbon emissions" OR "carbon footprint" OR LCA OR "life cycle assessment" OR "embodied energy" OR "embodied emissions"

AND

inventor* OR database OR EPD OR "environmental product declaration"

AND

disparity OR inconsisten* OR differ* OR varia OR compar* OR discrepanc* OR uncertaint* OR accuracy* OR review

AND

building OR construction OR architecture OR "built environment"

Searches of database records were restricted as follows:

EBSCO: abstract

Proquest: anywhere except full text

Scopus: title, abstract, keywords

Web of Science: topic and abstract

Results were filtered by the database options to those published between 1 January 2012 and 12 September 2022 as well as for Australia or an Australian geographical region.

Proquest (only) was additionally filtered to return articles only, as company reports were identified in the searches.

## Appendix B. Data Extraction Instrument

Each source was analysed by one or two of the reviewers to identify the following information:

- Source name
- Year of publication
- Authors names
- Are the authors involved in the creation of the carbon inventory they compare? If they are, which one?
- Is the dataset a known carbon inventory? If so, which one?
- List the materials compared.
- What was being compared?
- Which carbon calculation methods were included?
- What system boundary was used?
- What functional unit was used?
- Which carbon inventory was used?
- What values were compared?
- Was the source specifically about comparing carbon inventories or was it part of a larger study?
- What was the method of comparison?
- How were the results presented?
- What were the key findings?
- Was there any other discussion relevant to the research question?

A digital spreadsheet was used to record the above.

## Appendix C. Data for Figure 2

| | Wan Omar et al., 2014 [33] | Robati et al., 2016 [34] | Teh et al., 2017 [35] | Teh et al., 2018 [36] | Robati et al., 2019 [37] | Helal et al., 2020 [38] | Allende et al., 2020 [39] | Crawford et al., 2019 [40] | Crawford & Stephan 2020 [41] | Rodrigo et al., 2021 [18] | Robati & Oldfield 2022 [42] | Total # References Using This Material |
|---|---|---|---|---|---|---|---|---|---|---|---|---|
| aluminium | 1 | | | | 1 | | 1 | | 1 | | 1 | 5 |
| brass | | | | | | | 1 | | | | | 1 |
| bricks/blocks/pavers | 1 | | | | 1 | | | | 1 | | | 3 |
| carpet | | | | | 1 | | | | 1 | | 1 | 3 |
| cement | 1 | | | | | | | | 1 | | 1 | 3 |
| ceramic tile | | | | | | | 1 | | 1 | | 1 | 3 |
| concrete | 1 | 1 | 1 | 1 | | 1 | 1 | 1 | 1 | | 1 | 9 |
| copper | | | | | | | 1 | | 1 | | | 2 |
| fibre cement | | | | | | | | | 1 | | | 1 |
| glass | 1 | | | | | | 1 | | 1 | | 1 | 4 |
| insulation | 1 | | | | 1 | | 1 | | 1 | | 1 | 5 |
| mortar | | | | | | | | | 1 | | | 1 |
| paint | 1 | | | | | | 1 | | 1 | | 1 | 4 |
| pipes | 1 | | | | | | 1 | | 1 | 1 | | 4 |
| plasterboard | 1 | | | | 1 | | 1 | | 1 | | | 4 |
| plastics | | | | | 1 | | 1 | | 1 | | | 3 |
| roof tiles | | | | | | | | | 1 | | | 1 |
| sand | | | | | 1 | | 1 | | 1 | | | 3 |
| steel | 1 | | | | 1 | 1 | 1 | | 1 | | 1 | 6 |
| stone | | | | | | | 1 | | | | 1 | 2 |
| synthetic rubber | | | | | 1 | | | | | | | 1 |
| timber (milled and manufactured) | | | | | | | 1 | 1 | 1 | | 1 | 4 |

## Appendix D. Data for Figure 5

| Results Publication | Results Presented as | | | | | | |
|---|---|---|---|---|---|---|---|
| | Tables | Bar Chart | Line Chart | Box Plot | Scatter Plot | Probability Distribution Plot | % Differences |
| Wan Omer et al., 2014 [33] | 1 | 1 | | | | | |
| Robati et al., 2016 [34] | 1 | 1 | 1 | 1 | 1 | | |
| Teh et al., 2017 [35] | | 1 | | | | | |
| Teh et al., 2018 [36] | | 1 | | | | | |
| Crawford et al., 2019 [37] | | 1 | | | | | |
| Robati et al., 2019 [38] | 1 | | | 1 | 1 | 1 | |
| Helal et al., 2020 [39] | | | 1 | | | | |
| Allende et al., 2020 [40] | 1 | 1 | | | | | 1 |
| Crawford & Stephan 2020 [41] | | 1 | | | | | 1 |
| Rodrigo et al., 2021 [18] | 1 | | | | | | 1 |
| Robati & Oldfield 2022 [42] | | | 1 | 1 | | 1 | |

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
