# Peer review of "Embodied Carbon Inventories for the Australian Built Environment: A Scoping Review"

_buildings, doi:10.3390/buildings14030840_

Round 1

Reviewer 1 Report

Comments and Suggestions for Authors

The paper presents a scoping review about Embodied carbon inventories for the Australian built environment. It is a topic of interest to the researchers in the related areas, but the paper needs some improvements at this stage. My detailed comments are as follows:

1. The authors should add some more qualitative and timely references.

2. Figure 1 shows inconsistent arrow formatting. Please standardize the format and check for similar errors.

3. Why was the impact of regional differences in building materials not considered when evaluating the carbon values of different building materials? Please provide an explanation.

4. The article studied the carbon value of buildings, but did not expand on its significance, suggesting that articles on building energy management can be referred to, for example: DOI:10.17775/CSEEJPES.2021.04510.

5.The article mentions three methods for calculating carbon value, but does not compare them. Please explain the advantages and disadvantages of the three methods.

Comments on the Quality of English Language

Moderate editing of English language required

Author Response

Thank you for the time given for a thorough review.

Please find our responses in the attached word document.

Reviewer 2 Report

Comments and Suggestions for Authors

Manuscript ID: buildings-2854629

Manuscript Title: Embodied carbon inventories for the Australian built environment: a scoping review

The following corrections are recorded:

1.      The number of papers relied upon in the review is relatively small.

2.      Please give the reason for choosing C02eq instead of  C02e.

3.      The discussion needs to be improved.

4.      Remove the citation from the conclusion section.

5.      Please add the study limitations at the end of the conclusion.

Author Response

(The authors gave the same response as above.)

Reviewer 3 Report

Comments and Suggestions for Authors

The manuscript presents the scoping review on the topic of embodied carbon inventories for the Australian built environment. Considering the small scope and weak technical contents of the manuscript, it is not recommended for publication. Comments are provided as follows:

1.       The collected literature is up to September 2022, which shall be updated to reflect progress in this field in 2023.

2.       There are no quantitative discussions on how the embodied carbon differs from different sources. The paper discusses different life cycle stages coverage associated with the data and such discrepancy shall be addressed before the discussion on comparisons, which means that the author shall define the stages for the targeted comparison.

3.       There are some widely accepted standards to specify the calculation of embodied carbon, such as EN 15804, based on which the environmental product declarations were created. There is no such discussion in the manuscript.

4.       The manuscript mentioned both embodied carbon and embodied energy and discussed the appropriateness of their usage. To the reviewer, they are two indicators to address climate change and abiotic depletion potential impacts and are usually reported parallel in the EPD file.

5.       It is not sure why the review covers the format of the data presentation as shown in Figure 5. To the reviewer, the quantity of data is more important than its presenting format.

Some minor comments are provided as follows:

6.       Page 1, line 37. It is suggested to use subscript instead of superscript for “C02-eq”.

7.       Page 1, line 58, is there anything missing between the brackets?

8.       Page 16, line 494, shall it be “inventory” instead of “inventor”?

9.       Page 5, Table 1, Citations shall be added to the publications.

10.    Page 6. The readability of Table 2 shall be improved. It is suggested to enlarge the first column or add horizontal separators.

11.    Page 6, line 216. Please notice the additional “)”.

12.    Page 6, lines 216 and 217. It is suggested to provide some details about different methods besides their terms.

13.    It is suggested to review the format of the manuscript, especially those presented units.

Author Response

(The authors gave the same response as above.)

Round 2

Reviewer 1 Report

Comments and Suggestions for Authors

No other comments

Comments on the Quality of English Language

Minor editing of English language required

Author Response

Thank you

We have now completed a final proof reading of the document

Reviewer 3 Report

Comments and Suggestions for Authors

Thanks for the revision. The manuscript has been improved with more explanation and citations to the standards and recent publications. 

Author Response

Thank you very much for your feedback which has improved our paper